# Anti-Oxidative Effect of Pu-erh Tea in Animals Trails: A Systematic Review and Meta-Analysis

**DOI:** 10.3390/foods11091333

**Published:** 2022-05-04

**Authors:** Chiung-Ying Yang, Kuang-Chen Hung, Yea-Yin Yen, Hung-En Liao, Shou-Jen Lan, Hsin-Cheng Lin

**Affiliations:** 1Department of Healthcare Administration, Asia University, Taichung 413, Taiwan; l0921045376@gmail.com (C.-Y.Y.); heliao@asia.edu.tw (H.-E.L.); shoujenlan@gmail.com (S.-J.L.); 2Taichung Armed Forces General Hospital, Taichung 411, Taiwan; sur060@gmail.com; 3National Defense Medical Center, Taipei 114, Taiwan; 4Department of Oral Hygiene, College of Dental Medicine, Kaohsiung Medical University, Kaohsiung 807, Taiwan; yyyen0302@gmail.com

**Keywords:** anti-oxidative effect, pu-erh tea, systematic review, meta-analysis

## Abstract

This study adopted systematic literature review and meta-analysis methodology to explored anti-oxidative effect of pu-erh tea. Study authors have systemically searched seven databases up until 21 February 2020. In performing the literature search on the above-mentioned databases, the authors used keywords of pu-erh AND (superoxide dismutase OR glutathione peroxidase OR malondialdehyde). Results derived from meta-analyses showed statistically significant effects of pu-erh tea on reducing serum MDA levels (SMD, −4.19; 95% CI, −5.22 to −3.15; *p* < 0.001; I^2^ = 93.67%); increasing serum SOD levels (SMD, 2.41; 95% CI, 1.61 to 3.20; *p* < 0.001; I^2^ = 91.36%); and increasing serum GSH-Px levels (SMD, 4.23; 95% CI, 3.10 to 5.36; *p* < 0.001; I^2^ = 93.69%). Results from systematic review and meta-analyses validated that various ingredients found in pu-erh tea extracts had anti-oxidation effects, a long-held conventional wisdom with limited supporting evidence.

## 1. Introduction

In China, tea has long been regarded as a natural and healthy beverage with a long history of over 3000 years, and tea drinking has been gradually yet firmly imbedded into Chinese culture. Tea as a common kind of household beverage is widely favored because of its rich flavors, tastes, and biological activities. As determined by processing technologies employed and varying degrees of oxidation during manufacturing processes, Chinese teas are primarily divided into six broad categories: green tea (no oxidization), white tea (slightly oxidized), yellow tea (lightly oxidized), oolong tea (partially oxidized), black tea (fully oxidized), and dark tea (post-fermented) [1].

Due to their respective and distinct chemical components, teas of different types have their distinct flavors, quality characteristics, and associated health benefits. Among the many different types of tea, dark tea is especially unique in that it is made with post-fermentation, a processing technique commonly employed for more than 400 years, and the fermentation process is facilitated by a microorganism from a tea plant *Camellia sinensis var. sinensis* or *Camellia sinensis var. assamica,* belonging to the family Theaceae [2]. Dark teas can be further classified based upon different production areas and diverse processing technologies employed, including Yunnan pu-erh tea, Hunan Fu-zhuan tea, and others [3].

Pu-erh tea is generally regarded as the most popular dark tea and is widely consumed in southern parts of the Chinese mainland out of all different classifications of dark teas mentioned above. During the manufacture processes of dark tea, many microorganisms are used to participate in the post-fermentation, and *Aspergillus niger* is regarded as the dominant species [4].

The research found that tea polyphenols were oxidized to quinone and polymerized to form theaflavins, thearubigins, and theabrownins by microorganism action, which creates the specific taste and brownish-red color unique to dark tea [5,6].

Extracts of pu-erh tea can significantly lower plasma concentrations of glucose, insulin, triglycerides, and free fatty acids [7,8,9,10]. Possibly attracted by the myriad of potential health benefits associated with pu-erh tea, more people are taking up the habit of drinking pu-erh tea in recent years.

Recent studies have further confirmed that catechins, caffeine, polyphenols, amino acids, and polysaccharides in pu-erh tea extracts also have beneficial effects in decreasing atherosclerotic risk [11], weight reduction [12], and anti-hyperglycemic effect [13]. Pu-erh tea was presumed to have anti-hyperglycemic effects via inhibition on α-amylase and α-glucosidase. Yang et al. [14] used system review method to validate pu-erh tea’s inhibitory effects on α-glucosidase and α-amylase. Five English databases and three Chinese ones were searched up to 22 March 2018. The results showed pu-erh tea has significant inhibitory effects on α-glucosidase and α-amylase.

Several other studies [11,15,16] further examined the effects of pu-erh tea on blood antioxidant enzymes in a hyperlipidemia rat model. Compared to the hyperlipidemic control group, activities of superoxide dismutase (SOD) and glutathione peroxidase (GSH-Px) in serum were significantly elevated in pu-erh tea treated groups, while levels of malondiadehyde (MDA) markedly decreased. Results highlighted above indicate that pu-erh tea have strong anti-oxidative and lipid-lowering effects.

Jiang et al. [17] also established another hyperlipidemia rat model to examine anti-oxidative effects of pu-erh tea. The results show that the level of SOD significantly decreased in the treatment group, and levels of MDA and GSH-Px increased. The results are nevertheless different from those derived from the study of Hou et al. [11]. 

Chu et al. [18] explored the regulative efficacy of pu-erh tea extracts on human metabolic syndrome. Using randomized, double-blind, placebo-controlled method, results from their study indicated that subjects in intervention group administered with pu-erh tea extracts demonstrated excellent potential in improving central obesity, adjusting blood lipid, lowering blood sugar, regulating immunity, and resisting oxidation.

The precise mechanisms of anti-oxidative effects of pu-erh tea still remain unclear. As such, this study aimed to investigate the effects of pu-erh tea on anti-oxidation via both a systematic literature review and a meta-analysis method to analyze all relevant researches. This study also attempts to make comprehensive conclusions regarding the effectiveness of pu-erh tea on anti-oxidation.

## 2. Materials and Methods

To investigate and validate of pu-erh tea’s effect on anti-oxidation and other relevant factors, this study adopts the systemic literature review and meta-analysis methodology. This study was created in accordance to the “Preferred Reporting Items for Systematic Review and Meta-analysis Protocols” [19].

### 2.1. Information Sources and Searches

The study authors systemically searched the 7 databases of *Cochrane Library, EBSCO, PubMed, SCOPUS, Airiti Library, CNKI (China Knowledge Resource Integrated Database)*, and *Google Scholar* for potential studies to be considered for inclusion. As a great majority of pu-erh tea-related articles were written and published in Chinese, the authors also searched *Airiti Library* and *CNKI* databases for potentially eligible articles in the Chinese language. To further broaden the scope of literature search, *Google Scholar* was also researched to find other potentially eligible research articles written either in English or Chinese and which were not included in the traditional and academically more formal database aforementioned.

Yang C.Y., Yen Y.Y., and Lin H.C. were the 3 authors designated as the independent reviewers charged with reviewing quality and eligibility of the searched articles and further determining whether or not any specific searched literatures would be included or excluded from the study. In performing literature search on the above-mentioned databases, the authors used keywords of (Pu-erh OR Pu’er OR Pu-erh*) AND ((SOD) OR (superoxide dismutase) OR (GSH-Px) OR (glutathione peroxidase) OR (MDA) OR (malondialdehyde)).

### 2.2. Study Selection

As the first step of literature selection, the authors collectively determined to only include articles examined and determined qualified in meeting criteria of literature inclusion, as defined by PICO principle (population, interventions, compare, and outcome). The second step involved filtering out and excluding irrelevant articles based upon search criteria of research topic and abstract. The remaining full-text articles along with their respective abstract were then examined by the designated reviewers for eligibility for inclusion, per the inclusion criteria specified. The third step involved determining the ultimate eligibility for searched and screened articles to be included in meta-analysis, with only studies allocating mice to a pu-erh tea intervention versus a control group determined adequate and eligible for final inclusion. Exclusion criteria, defined as review articles, duplicate articles, non-full-text articles, articles that do not contain blood glucose changes, and non-animal trial studies, were also utilized by the authors to screen out ineligible articles.

### 2.3. Data Analysis

#### 2.3.1. Data Extraction

The included 9 articles were predominantly interventional studies, wherein pre-intervention and post-intervention data regarding changes in serum MDA, GSH-Px, and SOD were collected and measured in both experiment and control groups. Given that some of the included articles presented their study results in graphic forms only, and further considering specific strategies employed by individual studies in highlighting study results, the current study authors adopted below two methods to obtain research results:

As indicated by below formula, the first method involved calculating pooled standard deviation and average difference of MDA,SOD and GSH-Px values for the included studies, wherein pre-intervention and post-intervention average MDA,SOD and GSH-Px and standard deviation values were available.
$$S_{E_{-}dif}=\sqrt{\frac{\left(N_{E_{-}pos}-1\right){\left(SD_{E_{-}pos}\right)}^{2}+\left(N_{E_{-}pre}-1\right){\left(SD_{E_{-}pre}\right)}^{2}}{\left(N_{E_{-}pos}-1\right)\left(N_{E_{-}pre}-1\right)}}$$
$$d={\bar{X}}_{pos}-{\bar{X}}_{pre}\;S_{dif}=\sqrt{\frac{(N_{pos}-1)\left(SD_{pos}{)}^{2}+(N_{pre}-1\right){\left(SD_{pre}\right)}^{2}}{\left(N_{pos}-1\right)\left(N_{pre}-1\right)}}$$
where d is the average difference of the blood glucose value between pre-intervention and post-intervention with pu-erh tea in the trial group, while $S_{dif}$ is the standard deviation of the average value of blood glucose before and after the experiment.

For those studies included for analysis wherein blood glucose dates were presented only in graphic forms, the Screen Ruler Pro program tool (Accessed day: 5 March 2022; https://wonderwebware.com/screen-ruler-pro/) was adopted by the authors as the second method in measuring average values and the associated standard deviation related to participants’ blood glucose.

#### 2.3.2. The Effect Size

Traditionally, results from consolidation analysis were primarily expressed by a significance level combining method. Some argued that in order to better understand the size and significance of impact differences, researchers should also analyze the effect size (ES) of the included studies [20]. When compared with primary studies, “clarifying research methods” and “quantifying study results” were the two additional steps were required in meta-analysis studies. Effect size (ES) or standard common unit between various studies can be adequately determined by adopted the two additional steps aforementioned. When ES of individual studies was calculated, the overall average ES could be subsequently determined so as to measure and validate the intervention effect of all included studies. Standardizing the effect of averaging differences was necessary, as individual studies most likely involved the use of diverse scales, indexes, or tools for measurement. Of course, researchers may treat mean difference as a proper surrogate value for consolidation analysis and disregard the standardization processes if and when exactly the same scale, index, or tool for measurement was adopted by all individual studies involved, which is seldom the case.

### 2.4. Risk Bias and Quality Assessment

For experimental animal studies, systematic review (SR) approaches are not yet commonly adopted by researchers so far. However, awareness regarding potential merits associated with SR research methodology has gradually and steadily heightened among researchers. Because animal intervention studies are distinct from randomized clinical trials in numerous ways, researchers need to exercise necessary precaution and adapt their study strategy accordingly when attempting to employ SR research methodology of clinical trials onto animal intervention studies in order to achieve the optimal research outcomes.

SYRCLE’s RoB tool, a risk-of-bias tool specifically for animal intervention studies, which found its root from the Cochrane risk of bias tool and has been amended to accommodate for bias aspects having a unique role in animal intervention studies, was adopted by Hooijmans et al. [21].

There are 10 bias-related entries included in the RoB tool suitable for animal studies. These 10 entries cover potential aspects of bias related to selection, performance, detection, attrition, and reporting, among others. In total, 50% of items included in the RoB tool share identical characteristics with items used in the conventional Cochrane RoB tool, with design differences between animal studies and RCTs remaining primary variations between such two risk-of-bias tools. The fact that some researchers do not fully understand some unique aspects of experimental design related to animal studies as compared with those aspects related to clinical studies, coupled with some limitations or shortcoming of RoB tool itself, may to a certain degree contribute to the abovementioned variations. RoB tool adopted by SYRCLE is considered an adaptive version tracing its root to the traditional Cochrane RoB tool. To facilitate, refine, and optimize critical evaluation of evidences gleaned from animal studies, more widespread utilization of this risk of bias tool is required. With widespread and more frequent utilizations of the RoB tool, researchers can subsequently be assured of the efficiency and effectiveness of the RoB tool in translating animal studies onto clinical practices. Furthermore, researchers may also increase their awareness regarding the need to enhance methodological quality of animal studies.

Per the Cochrane Handbook for Systematic Reviews of Interventions [22], two authors (Yang C.C., Lin H.C.) assessed the presence of high, low, or unclear risk of bias of studies included for systematic analyses. Generation of random sequence, participants and research staff blinding, allocation concealment, outcome assessment blinding, outcome data considered incomplete, and outcome reporting were items included for assessment by study authors.

### 2.5. Analysis

In analyzing the included studies’ high heterogeneity potentially caused by a random effect, the Comprehensive Meta-Analysis 2.2.064 was utilized by current study authors. In measuring effect size (ES), SMD and 95% confidence intervals (CIs) were adopted. Forest plot was then employed by study researchers in presenting the weighted mean difference (WMD) and the 95% confidence interval (95% CI). Subsequently, the statistic report of Begg’s test was also compiled by the study authors.

## 3. Results

### 3.1. Search Results

The first-stage literature search up until 21 February 2020 retrieved 208 research articles from database searches (52 English articles and 156 Chinese ones). Among the 52 articles written in English, 17 were retrieved from *PubMed*, 13 from *EBSCO*, 21 from *SCOPUS*, and 1 from *Cochrane Library*. As for the 156 articles written in Chinese during the first stage search, 14 were retrieved form *Airti* Library, 82 from *China National Knowledge Infrastructure (CNKI)*, and 60 from *Google Scholar*. The complete search process is illustrated in the PRISMA flow chart (Figure 1) [19].

The second stage removed 29 duplicates and review articles considered unsuitable for meta-analysis, with 179 articles remaining. After scanning abstracts and reading study tiles, 167 articles were considered inappropriate for meta-analysis and were further screened out, leaving 12 articles with potential eligibility for inclusion, as shown in the third stage.

As indicated in the fourth stage of literature search, none were excluded from the 12 remaining articles on grounds of being non-full-text. The fifth stage involved screening out articles for not meeting PICO criteria, and all 12 articles were determined eligible. In the sixth and final stage, three studies in mice were further eliminated from inclusion, leaving a total of nine articles included for meta-analyses (as presented in Table 1).

### 3.2. Quality Assessment of Selected Studies

In assessing risk of bias of the nine in-vivo studies included for meta-analyses, SYRCLE’s risk-of-bias (RoB) tool, purpose-built for animal studies (Hooijmans, Rovers et al., 2014), was adopted by the current study. Findings derived from using the SYRCLE’s risk-of-bias (RoB) tool in assessing quality of each individual trial by answering 10 questions listed, expressed as Yes, unclear, of No, were compiled by the authors and shown as Table 2. Two authors (C.Y. Yang and H.C. Lin) were tasked with the responsibility of using the SYRCLE’s risk-of-bias (RoB) tool in independently to assess the risk of bias of the included studies.

### 3.3. A Systematic Review of the Literature

Adopting pre-established inclusion and exclusion criteria in assessing eligibility of the searched articles for inclusion, nine studies were collectively determined to have met all criteria and then included for systematic review and meta-analyses. As indicated in Table 1, main observation indexes included types of animals, experimental model, intervention duration, and outcome.

Using a rat hyperlipidemia model, Hou et al. [11] conducted a study to investigate pu-erh’s effects on weight gain, serum levels of lipids and lipoproteins, lipid oxidation, and blood antioxidant enzymes. In their trial, water extracts of pu-erh tea of 0.5, 1.5, or 3.0 mg/kg were given to the rats. Activities of SOD and GSH-Px in serum were markedly elevated for rats in experiment group receiving pu-erh tea treatment as compared with those of rats in the hyperlipidemic control group. Furthermore, levels of MAD (a by-product of lipid peroxidation) decreased substantially for rats in the same (pu-erh treatment) group. Rats given the highest dose of extracts from fermented pu-erh tea in the treatment group were found to have the most pronounced effects as indicated above. Results aforementioned have validated pu-erh tea’s potent anti-oxidative effects.

In a similar study also using a hyperlipidemia rats model, Jiang et al. (2009) [17] investigated anti-oxidation effects of pu-erh tea, oolong tea, and herbs, respectively. Findings derived from their study showed significant reductions of SOD in rats of the treatment group. The same study also found substantial increases of serum MDA and serum GSH-Px for rats given pu-erh tea, oolong tea, and herbs in the treatment group.

Another study was conducted by Hou et al. [23] to investigate the varying effects on level and peroxidation of serum lipids of experimental hyperlipidemia rats fed with pu-erh teas made with distinct processing techniques. Results from their study showed substantial increases of SOD and GSH-Px activities and, at the same time, significant decreases of serum MDA contents for rats in the treatment group. Their study showed various degrees of anti-oxidative and anti-hyperlipidemia effects exhibited by pu-erh teas processed with distinct techniques. Pu-erh teas made with different processing techniques were all found to have contributed to lowering atherosclerosis risk of study animals although to various degrees.

By feeding pu-erh tea extracts of distinct doses to hyperlipidemia model rats for a duration of 35 days, Xu et al. [24] conducted a trial to investigate the effects and possible mechanisms of pu-erh tea on preventing hyperlipidemia, protecting endothelium of blood vessels, and contributing to anti-oxidation. By comparing results from rats fed with fermenting pu-erh tea and fermented pu-erh tea in the treatment group and that from rats in the control (hyperlipidemia model) group, their study found that for rats in treatment (pu-erh tea) group, SOD and GSH-Px activities elevated substantially, while MDA contents significantly decreased. Findings from their study further revealed that the effects aforementioned were more pronounced for rats administered with fermented pu-erh tea than those of their counterpart rats fed with fermenting pu-erh tea.

Liu et al. [15] conducted a study aimed at investigating hepatocyte cytochrome expression in a non-alcoholic steatosis (fatty liver) rats model. In their trial, all study animals were sacrificed under anesthesia 35 days after the initiation of experiment, and levels of serum MDA, GSH-Px, SOD transaminase activities, and liver pathological changes were observed and recorded by researchers. In contrast with results from rats in hyperlipidemia model group, levels of serum MDA were reduced substantially in rats administered with low-dose pu-erh tea. Their study further revealed that fermented pu-erh tea’s effects in improving enzyme activities of GSH-Px and SOD were more pronounced than those of black tea and Tieguanyin (tea).

In a study conducted by Q. Wang et al. [25], researchers aimed to explore the effects of anti-oxidation by large molecular polymeric pigments (LMPPs) extracted from Zijuan pu-erh. Rats fed with extracts of Zijuan pu-erh were found to have had much higher levels of serum SOD and GSH-Px activities and substantially lower levels of serum MDA than their counterparts in the hyperlipidemia model group. In their study, the concentrations of serum SOD and GSH-Px activities were found to have increased by 66.88% and 29.09%, respectively. On the other hand, concentrations of serum MDA were found to have decreased by 59.11% in rats receiving high-dose pu-erh tea extracts as compared with those of rats in control (hyperlipidemia model) group. Results of their study validated that extracts from pu-erh tea exhibit good anti-oxidation effects, and as such, pu-erh teas should be regarded as a source of natural antioxidants.

Fermented pu-erh tea’s effects on lipid peroxidation in rats with fatty liver were explored in a study conducted by R. Wang et al. [16]. Results from their study indicated that levels of serum SOD and GSH-Px were elevated more substantially in rats administered with medium-dose and high-dose extracts from fermented pu-erh tea than those of rats in the positive control group. Their research further showed significant elevation of liver GSH-Px in rats receiving high-dose extracts of fermented pu-erh tea and substantial increase of serum DOD levels in rats fed with either medium-dose or high-dose extracts from fermented pu-erh tea. Significant reductions in serum MDA levels were observed in rats administered with either low, medium, or high doses of fermented pu-erh tea extracts, while obvious increases of liver tissue MDA levels were noted in rats receiving high-dose fermented pu-erh extracts.

Pu-erh tea extracts’ effects on decreasing oxidative stress and subsequently facilitating hepatic protection (liver protection) in rats given a high-fat diet were explored by Su et al. [26]. Results from their study showed that extracts from pu-erh tea could contribute to reductions of serum MDA levels.

In a study conducted by Zheng et al. [27], researchers investigated the anti-oxidative effects of Ganpu tea, an increasingly popular tea drink manufactured from pu-erh and pericarp from Citrus reticulata “Chachi (Guangchenpi)”. Their study found that Ganpu tea and GCP could elevate the activities of SOD by 13.4 % and 15.1% and increase GSH-Px activities by 16.3% and 20.5%, respectively.

### 3.4. Meta-Analysis

Authors of the current study conducted meta-analyses by using data derived from the nine included articles [11,15,17,23,24,25,26,27,28]. Serum levels of MDA, SOD, and GSH-Px were selected to be the end points of study outcomes.

#### 3.4.1. Heterogeneity Test and Meta-Analysis Results

##### Serum MDA

For the end point of serum MDA, the authors analyzed 28 datasets derived from the 9 included studies, using Comprehensive Meta-Analysis Software (CMA) to investigate whether or not the time-to-attendance was impacted and whether or not there was heterogeneity. As indicated in Figure 2, statistical results were noted to be highly heterogeneous, with study findings showing that pu-erh’s effects on reducing serum MDA levels were statistically significant (SMD, −4.19; 95% CI, −5.22 to −3.15; *p* < 0.001; I^2^ = 93.67%).

##### Serum SOD

For the end point of serum SOD, the authors analyzed 25 datasets from 8 articles included for analyses, also using the CMA software to investigate whether or not time-to-attendance was impacted and whether or not there was heterogeneity. As shown in Figure 3, statistical results were also found to be highly heterogeneous, indicating that pu-erh’s effects on increasing serum SOD levels were statistically significant (SMD, 2.41; 95% CI, 1.61 to 3.20; *p* < 0.001; I^2^ = 91.36%).

##### Serum GSH-Px

For the end point of serum GSH-Px, the authors analyzed 25 datasets from 8 articles included and, using the CMA software, investigated whether or not time-to-attendance was impacted and whether or not there was heterogeneity. As noted in Figure 4, statistical results were also revealed to be highly heterogeneous, indicating pu-erh’s effects on increasing serum GSH-Px levels were determined to be statistically significant (SMD, 4.23; 95% CI, 3.10 to 5.36; *p* < 0.001; I^2^ = 93.69%).

## 4. Discussion

Malondialdehyde (MDA) is the organic compound with the nominal formula CH2 (CHO) 2. A study by Gawel et al. [29] noted that malondialdehyde (MDA) is one of the final end products of polyunsaturated fatty acids peroxidation in the cells. An increase of free radicals tends to cause MDA overproduction. MDA level is also popularly identified as a surrogate marker for oxidative stress as well as for antioxidant status among patients with cancer, based on reasons abovementioned.

A study conducted by Younus [30] found that SOD was one of the antioxidant defense mechanisms to fend off oxidative stress in human body. SOD was also revealed to have an excellent therapeutic effect for patients with reactive oxygen species-mediated conditions. SOD’s therapeutic effects were noted to exist in different physiological and pathological conditions, including cancer, inflammatory diseases, cystic fibrosis, ischemia, aging, rheumatoid arthritis (RA), neurodegenerative diseases, and diabetes.

Lubos et al. [31] indicated that reactive oxygen species, such as superoxide and hydrogen peroxide, are generated in all cells by mitochondrial and enzymatic sources. These reactive species can cause oxidative damage to DNA, proteins, and membrane lipids. GSH-Px is an intracellular antioxidant enzyme that reduces hydrogen peroxide to water to limit its harmful effects. By reducing hydrogen peroxide accumulation, GSH-Px therefore could also modulate these processes.

In contrast with conclusions drawn from other studies listed in Table 1 for systemic review, results from study by Zheng et al. [27] indicated that pu-erh’s effects on lowering MDA levels were statistically non-significant. In their study, Jiang et al. [17] even reported that pu-erh would enhance the levels of serum MDA while reducing the levels of serum SOD. Potentially conflicting findings listed above drove the authors of the current study to further investigate the anti-oxidative effects of pu-erh tea.

Findings derived from meta-analyses of current study indicated that pu-erh tea could enhance serum levels of both SOD (SMD, 2.41; 95% CI, 1.61 to 3.20; *p* < 0.001; I^2^ = 91.36%) and GSH-Px (SMD, 4.23; 95% CI, 3.10 to 5.36; *p* < 0.001; I^2^ = 93.69%) on the one hand and reduce levels of serum MDA (SMD, −4.19; 95% CI, −5.22 to −3.15; *p* < 0.001; I^2^ = 93.67%) on the other hand.

Chu et al. [18] explored the regulative efficacy of pu-erh tea extract on metabolic syndrome. The results showed that pu-erh tea could markedly reduce MDA level (5.10–6.77 nmol/mL) and enhance SOD level (81.17–100.95 U/mL). Based upon findings from meta-analyses of the current study, results related to humans were found to be similar.

When conducting systematic review, findings of most the included studies supported that anti-oxidation effects of pu-erh tea were statistically significant. However, it is noteworthy that many experiment design-related factors, including but not limited to model of experiment, PTPS intervention concentration and duration of intervention, may influence the ultimate results of individual studies. Results from a study conducted by Zheng et al. [27] further reported that pu-erh’s effect was limited to enhancing GSH-Px levels only. As Zheng et al. did not use hyperlipidemia model in their study to validate pu-erh tea’s effects, an experiment design commonly adopted by all other included studies, the authors of the current study suspected that different experiment design may have contributed to distinct research findings.

Zhang et al. [32] reported that anti-oxidative activities of phenolic components extracted from pu-erh were much potent than those of vitamin C, further implicating that pu-erh may be an ideal natural antioxidant readily available. Among the aforementioned phenolic components, gallic acid, (+)-catechin, (−)-epigallocatechin-3-O-gallate, (−)-epiafzelechin-3-O-gallate, kaempferol, and quercetin were richer than those of other components.

## 5. Conclusions

Results from systematic review conducted in the current study validated that various ingredients found in extracts of pu-erh had anti-oxidation effects, a long-held conventional wisdom with limited supporting evidences. A study on metabolic syndrome by Chu et al. (2011) also drew similar conclusion. Notwithstanding the aforementioned, no studies have attempted to explore individual ingredients behind pu-erh’s anti-oxidation effects and investigate their respective mechanisms at work.

The number of research articles included in systematic review and subsequent meta-analyses was constrained by two reasons. Firstly, though pu-erh tea is quite popular in Chinese-speaking communities, it is much less known by and accessible to Westerners, and consequently, there are relatively few researches conducted by researchers from English-speaking countries. Secondly, distinct data formats and different research designs in many of the searched articles have rendered their data either unusable or inadequate for subsequent meta-analyses.

By reviewing and subsequently analyzing data sets contained within the nine studies included, findings from the current study validated the anti-oxidation effects of pu-erh, an ingrained claim from tea drinkers but with limited evidences.

None of those articles included, reviewed, and subsequently analyzed have attempted to investigate and then pinpoint specific components or ingredients behind pu-erh’s anti-oxidation effects, ideal concentration for individual component, or optimal intervention duration. As such, the authors of the current study are of the opinion that more studies are warranted to more conclusively answer those important questions. Lv et al. [33] indicated that Pu-erh tea had antioxidant activities, and there were significant positive correlation with the levels of EGCG, the levels of total flavonoids, and theabrownins. Guo et al. [34] indicated that pu-erh tea polysaccharides exhibited obvious antioxidant activities. Moreover, the heat map analysis found that total phenolic and protein contents in pu-erh tea polysaccharides were positively correlated with their antioxidant, indicating that the presence of phenolic compounds and proteins in the pu-erh tea polysaccharides might be the main contributors to their bioactivities. Perhaps tea phenol components’ effects on anti-oxidation would be an appropriate starting point in the near future for interested researchers.

## Figures and Tables

**Figure 1 foods-11-01333-f001:**
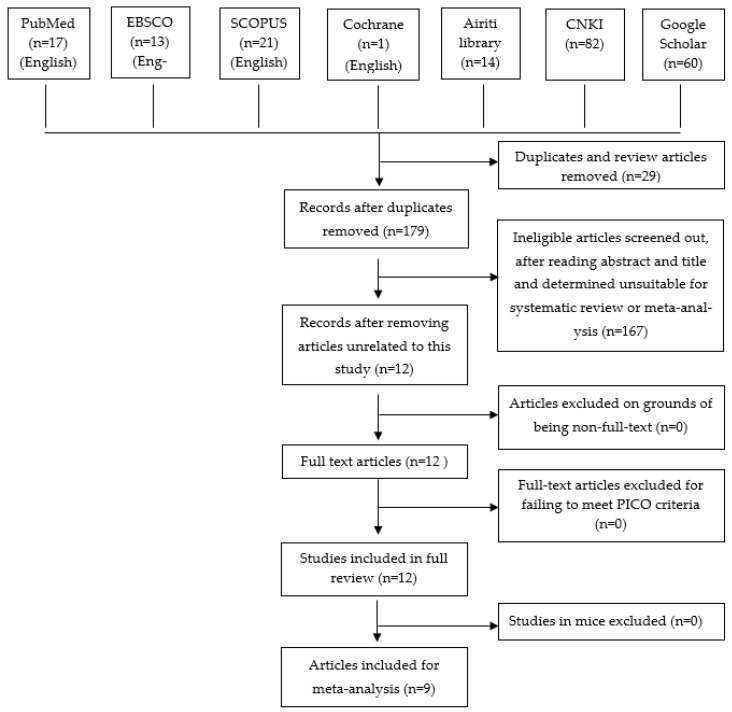
Flowchart of search results and article retrieval.

**Figure 2 foods-11-01333-f002:**
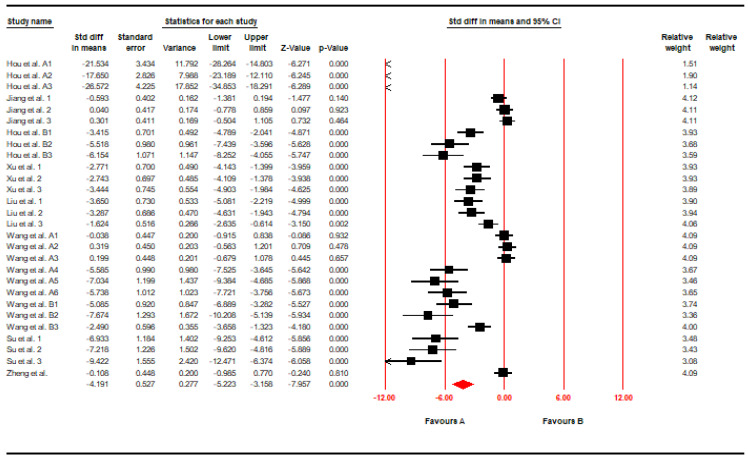
Forest plot of decreasing Serum MDA (SMD, −4.19; 95% CI, −5.22 to −3.15; *p* < 0.001; I^2^ = 93.67%). Black solid square indicated the individual study’s SMD and the red solid square indicated the polling SMD of the included studies.

**Figure 3 foods-11-01333-f003:**
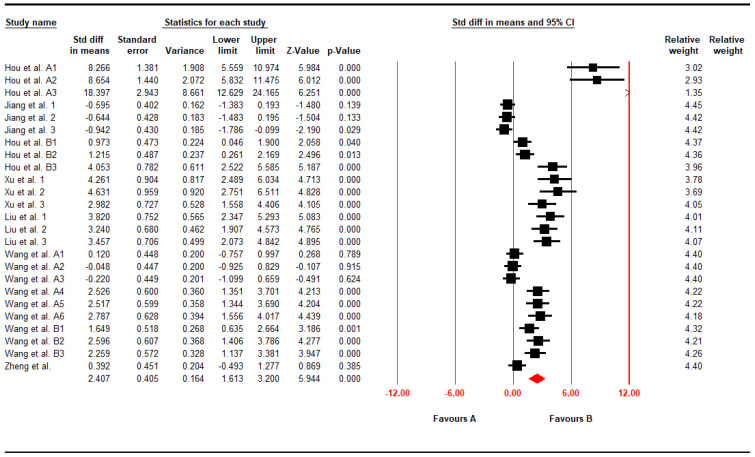
Forest plot of Elevating Serum SOD (SMD, 2.41; 95% CI, 1.61 to 3.20; *p* < 0.001; I^2^ = 91.36%). Black solid square indicated the individual study’s SMD and the red solid square indicated the polling SMD of the included studies.

**Figure 4 foods-11-01333-f004:**
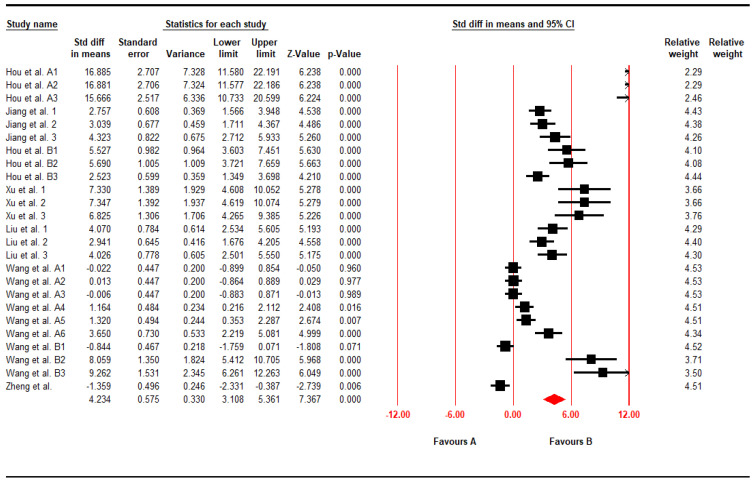
Forest plot of Elevating Serum GSH-Px (SMD, 4.23; 95% CI, 3.10 to 5.36; *p* < 0.001; I^2^ = 93.69%). Black solid square indicated the individual study’s SMD and the red solid square indicated the polling SMD of the included studies.

**Table 1 foods-11-01333-t001:** System review of pu-erh tea anti-oxidative effect in rats.

Study	Animals (Experimental Model)	Intervention	Duration (Endpoint)	Outcome
Hou, Shao, et al. (2009) (China) [11]	Male SD Rats (Hyperlipidemia model)	Rats were treated with water extracts of fermented or unfermented pu-erh tea.	30 days (Serum: MDA, SOD, and GSH-Px)	Compared to the hyperlipidemic control group, activities of SOD and GSH-Px in serum were significantly elevated in pu-erh tea-treated groups, while levels of MAD decreased in the same groups. These effects were most pronounced in the groups treated with the highest dose of fermented pu-erh tea extract. The study results suggest that pu-erh tea exerts strong antioxidative and lipid-lowering effects and therefore can be used to reduce the risk of cardiovascular disorders.
Jiang et al. (2009) (China) [17]	Male SD Rats (Hyperlipidemia model)	Rats were fed with high-lipid diet, oolong tea, herb, and fermented and unfermented pu-erh tea.	35 days (Serum: MDA, SOD, and GSH-Px)	The result indicated that SOD content reduced significantly in treatment group, while MDA and GSH-Px increased. The result concluded that the herb, oolong tea, and pu-erh tea are able to regulate the level of serum anti-oxidation.
Hou, Xiao, et al. (2009) (China) [23]	Male SD Rats (Hyperlipidemia model)	Rats were treated with water extracts fermented or unfermented pu-erh tea.	30 days (Serum: MDA, SOD, and GSH-Px)	The activities of SOD and GSH-Px increased, with the contents of MDA in serum markedly decreased. Pu-erh tea processed with different techniques has effects of anti-oxidative and anti-hyperlipidemia and contributes to reduce the risk of atherosclerosis.
Xu et al. (2010) (China) [24]	Male SD Rats (Hyperlipidemia model)	Rats were fed with high-lipid diet, oolong tea, and water extracts of fermented and unfermented pu-erh tea.	35 days (Serum: MDA, SOD and GSH-Px)	At experimental endpoint, blood lipids levels, activities of SOD, GSH-Px, and MDA in serum were measured. The comparison among fermenting and fermented pu-erh tea groups and model group revealed that activities of SOD and GSH-Px increased significantly; however, the MDA contents decreased in pu-erh tea groups. The effect of fermented pu-erh tea was more significant than fermenting pu-erh tea.
Liu et al. (2013) (China) [15]	Wistar Rats (Hyperlipidemia model)	Rats were divided into normal group, hyperlipidemia model group, fermented pu-erh tea group, Tieguanyin group, and low-, medium-, and high-dose black tea group, respectively.	35 days (Serum: MDA, SOD and GSH-Px)	The aim was to detect the expression of hepatocyte cytochrome in rat non-alcoholic steatosis model. All rats were sacrificed after 35 days’ feeding; serum levels of MDA, GSH-Px, and SOD transaminase activity and pathological changes of the liver were observed. RT-PCR method was used to test expression quantity in rat cells. Compared to the hyperlipidemia model group, the serum levels of MDA in low-dose group of the fermented pu-erh tea were significantly decreased (<0.05); fermented pu-erh Tea had a better effect on improving enzyme activity of GSH-Px and SOD than those of Tieguanyin and black tea.
Q. Wang et al. (2013) (China) [25]	Male SD Rats (Hyperlipidemia model)	Rats were divided into 6 groups,: a normal control group; a hyperlipidemia model group; and low-, medium-, and high-dose treatment group.	7 days (Serum: MDA, SOD, and GSH-Px)	In vivo, the extract from Zijuan pu-erh tea-treated rat groups showed significantly increased serum SOD and GSH-Px activities and reduced MDA in the hyperlipidemia model group. The serum SOD and GSH-Px activities concentration were 66.88 and 29.09 higher, respectively, whereas the serum MDA concentrations were 59.11% lower in the high-dose treatment group than in the hyperlipidemia model group. These results showed that pu-erh extract has a good anti-oxidative function and can be considered as a natural antioxidant source.
R. Wang et al. (2013) (China) [16]	Wistar Rats (Hyperlipidemia model)	Rats were divided into 6 groups: a normal control group; a hyperlipidemia model group; and low-, medium-, and high-dose treatment group.	60 days (Serum: MDA, SOD, and GSH-Px; Liver: MDA, SOD, and GSH-Px)	The study investigated the effects of fermented pu-erh tea on lipid peroxidation in alcoholic fatty liver rats. The result showed that compared to the positive control group, the serum levels of SOD and GSH-Px in medium- and high-dose groups of the fermented pu-erh tea are significantly increased. The liver GSH-Px in high-dose group of fermented pu-erh tea and SOD in medium- and high-dose groups of the fermented pu-erh tea are significantly increased. The serum levels of MDA in low-, medium-, and high-dose group of the fermented pu-erh tea group and the liver tissue level of MDA in high-dose group of the fermented pu-erh tea are significantly decreased.
Su et al. (2016) (China) [26]	SD Rats (Hyperlipidemia model)	Fifty SD rats were divided into five groups: a normal control group; a hyperlipidemia model group; and low-, medium-, and high-dose treatment group.	12 weeks (Serum: MDA)	Reducing oxidative stress and hepatoprotective effect of Pu-erh tea water extracts on rats fed with high-fat diet were researched for explaining health care of Pu-erh tea. The result demonstrated that Pu-erh extract caused the decreases MDA levels, and the increases in hepatic SOD and GSH-Px activities, indicating that the extract may be reducing oxidant stress state and inhibiting lipid peroxidation, thus decreasing the activities of ALT and AST, and protecting the liver in rat.
Zheng et al. (2020) (China) [27]	SD Rats (none)	The rats were randomly assigned into 4 groups: Control group; GTE group (0.2 g/mL, 15 mL/kg/d); PTE group (0.16 g/mL, 15 mL/kg/d); and GCPE group (0.04 g/mL, 15 mL/kg/d).	28 days (Serum: MDA, SOD, and GSH-Px)	Ganpu tea is an emerging tea drink produced from pu-erh tea and the pericarp of *Citrus reticulata* “Chachi (Guangchenpi)”. Recently, it has been increasingly favored by consumers due to the potential health effects and special taste. Ganpu tea and GCP could significantly enhance the activities of SOD by 13.4% and 15.1% as well as the activities of GSH-Px by 16.3% and 20.5%, respectively.

**Table 2 foods-11-01333-t002:** SYRCLE’s Risk-of-Bias Tool.

SYRCLE’s Risk of Bias Tool for Animal Studies	Hou et.al., 2009 A [11]	Jiang et al., 2009 [17]	Hou et al., 2009 B [23]	Xu et al., 2010 [24]	Liu et al., 2013 [15]	Wang et al., 2013 A [25]	Wang et al., 2013 B [16]	Su et al., 2016 [26]	Zheng et al., 2020 [27]
1. Was the allocation sequence adequately generated and applied?	Yes	Yes	Yes	Yes	Yes	Yes	Yes	Yes	Yes
2. Were the groups similar at baseline, or were they adjusted for confounders in the analysis?	Yes	Yes	Yes	Yes	Yes	Yes	Yes	Yes	Yes
3. Was the allocation adequately concealed?	Unclear	Unclear	Unclear	Unclear	Unclear	Unclear	Unclear	Unclear	Unclear
4. Were the animals randomly housed during the experiment?	Unclear	Yes	Unclear	Yes	Yes	Yes	Yes	Yes	Yes
5. Were the caregivers and/or investigators blinded from knowledge of which intervention each animal received during the experiment?	Unclear	Unclear	Unclear	Unclear	Unclear	Unclear	Unclear	Unclear	Unclear
6. Were animals selected at random for outcome assessment?	No	No	No	No	No	No	No	No	No
7. Was the outcome assessor blinded?	Unclear	Unclear	Unclear	Unclear	Unclear	Unclear	Unclear	Unclear	Unclear
8. Were incomplete outcome data adequately addressed?	No	Yes	No	Unclear	Yes	Yes	Yes	Yes	Yes
9. Are reports of the study free of selective outcome reporting?	Unclear	Unclear	Unclear	Yes	Yes	Unclear	Yes	Yes	Yes
10. Was the study apparently free of other problems that could result in high risk of bias?	Unclear	Unclear	Unclear	No	No	Unclear	No	No	No

## Data Availability

The data presented in this study are available on request from the corresponding author.

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
