# Peer review of "Anti-Oxidative Effect of Pu-erh Tea in Animals Trails: A Systematic Review and Meta-Analysis"

_foods, 2022, doi:10.3390/foods11091333_

Round 1

Reviewer 1 Report

This manuscript is a review of the antioxidative effects of Puer tea.

There are some changes authors need to make, for example:

Line 30 italics for Camelia sinensis

Line 68 Yang et al. (2019)[14]

Line 291 Authors need a reference to Figure 3 after “PRISMA flow chart”

Authors should review the instructions for authors on how to cite references in the text. They are not using a uniform format. Similarly, the authors must review the references one by one in the list of references. In some cases, they use the full name of the journal; in other cases, they use the abbreviated form.

In the case of Table 1, they should also review the way of citing the references.

The introductory section is not clear why they include sections 1.2 (1.2.1 and 1.2.2) since it is only a very general description of reactive oxygen species and endogenous antioxidant mechanisms. Still, they do not relate to tea intake.

The sequence of section 1.2.4 is also unclear because they include it here instead of a separate section, eg 1.3.

I consider that the introduction's discussion part is a repetition since they only cite references on tea's antioxidant capacity.

The conclusions section should be improved; they repeat that tea has an antioxidant capacity and that no studies refer to individual tea compounds' antioxidant capacity.

Reviewer 2 Report

Dear Authors,

Firstly I would like to congratulate you on an attempt to discuss such an important topic and to present the findings. I had a pleasure in reading this manuscript and please find below some of my comments/suggestions provided in Minor section. Well done to the team. I sincerely hope that this comments/suggestions assist in the improvements of the manuscript.

Minor:

  1. In several places, please use full name then followed by the abbreviation.
  2. Authors should consider changing the title to indicate that the manuscript is relating to animal trials.
  3. Please use italics for all species and genera names
  4. In several places, authors should use consistent language Alpha-amylase vs α-amylase
  5. Authors should consider deleting line 54-58 as it does not bring much to the manuscript overall.
  6. Line 491 – 492 – can authors double check the values included.
  7. Authors should consider modifying the text in several places not to over-extrapolate the findings observed in animal trials with the potential findings in human studies.
  8. Reference list should be consistent – please revisit
